# Emerging unprecedented lake ice loss in climate change projections

Lei Huang [1,2] ✉, Axel Timmermann [1,2], Sun-Seon Lee [1,2], Keith B. Rodgers [1,2], Ryohei Yamaguchi [1,2,3] & Eui-Seok Chung [4]

Seasonal ice in lakes plays an important role for local communities and lake ecosystems. Here we use Large Ensemble simulations conducted with the Community Earth System Model version 2, which includes a lake simulator, to quantify the response of lake ice to greenhouse warming and to determine emergence patterns of anthropogenic lake ice loss. Our model simulations show that the average duration of ice coverage and maximum ice thickness are projected to decrease over the next 80 years by 38 days and 0.23 m, respectively. In the Canadian Arctic, lake ice loss is accelerated by the cold-season polar amplification. Lake ice on the Tibetan Plateau decreases rapidly due to a combination of strong insolation forcing and ice-albedo feedbacks. Comparing the anthropogenic signal with natural variability represented by the Large Ensemble, we find that lake ecosystems in these regions may be exposed to no-analogue ice coverage within the next 4-5 decades.

More than 58.5 million lakes on Earth are covered by seasonal and perennial lake ice[1,2]. Over recent decades a widespread loss of lake ice has been reported[3–6], with many lakes experiencing delayed winter freezing or an earlier spring melt. Offline-lake model simulations forced with the output obtained from multi-model climate simulations[7] were able to link the observed lake ice loss to greenhouse warming. Moreover, it was found that the global reduction of lake ice will further intensify in response to projected future warming[2,7,8].

Other studies that address lake ice responses to greenhouse warming were conducted with statistical models, which are built on the empirical relationship between the observational ice phenology and meteorological variables[2,9]. Their statistical projections document that a large number of lakes are likely to experience intermittent or permanent winter ice cover loss in the Northern Hemisphere until the end of this century, but they do not provide projection of changes in thickness and duration of ice cover. Physical process-based lake-ice model simulations, on the other hand, have been used in recent years[7,8,10] to assess future climate change in lake systems. They run as offline models which are typically forced by daily meteorological data derived from climate model simulations. While thermodynamic processes of water and ice are resolved

explicitly in these simulations, the three-way interaction between lake, lake ice and atmosphere, is not represented. Such coupling will play an important role in particular for large lake systems which are known to have considerable impacts on local climate[11–13]. Another caveat in such offline model simulations is the fact that diurnal cycle thermodynamics are not resolved. We address these limitations here by including lake simulators into Earth system models as an interactive component[14,15].

Lake ice has experienced unprecedented loss rates in the last decades[16], and lake organisms are suffering severe changes in their habitat with the shortening of ice duration which had led to reorganization of aquatic ecosystems, for example, in circumpolar lakes[17]. Exceeding climate-related thresholds of lake ice phenology can even accelerate such biological processes[18]. Lake organisms may be exposed to no-analog ice conditions in the future, which are unlikely under natural climates. What's more, disappearance of extant ice conditions will increase the risk of local extirpation of populations for cold species[19]. It is thus fundamental to project time and the corresponding global warming levels for the emergence of no-analog conditions for risk assessment and adaptation in the future. Here, we set out to provide a mechanistic account for the sensitivity of lake systems to greenhouse warming, as well as to establish

[1]Center for Climate Physics, Institute for Basic Science, Busan, South Korea. [2]Pusan National University, Busan, South Korea. [3]Japan Agency for Marine-Earth Science and Technology, Yokosuka, Japan. [4]Korea Polar Research Institute, Incheon, South Korea. ✉e-mail: huanglei@pusan.ac.kr

emergence timescales for anthropogenic signals above the natural noise background with a fully coupled Earth system model that includes an interactive lake simulator. The lake model used here represents thermodynamic processes including ice growth and melt, and vertical mixing[20] and is coupled to the atmosphere through the exchange of heat and momentum.

Detecting the human influences on lake ice phenology and estimating when the anthropogenic signal will emerge from the natural background noise is contingent on knowing the degree of natural variability of the regional lake ice fluctuations. Previous studies have focused on multi-model ensemble forcing approaches, which typically result in a suppression of natural variability and mix the sensitivity from different climate models[21,22]. Therefore, when no-analog conditions of lake ice will emerge still remains to be answered. Here we pursue a different modeling approach: we focus on only one Earth system model, which has been run for the historical period and a future anthropogenic emission forcing scenario 100 times. These so-called large ensemble simulations are obtained by perturbing the initial conditions. At every point in time, we therefore have a physically consistent single-model-based estimate of the anthropogenic signal (ensemble mean) and of natural variability (ensemble spread). This is advantageous when calculating the time of emergence of anthropogenic signals and the emergence of no-analog conditions for aquatic ecosystems, relative to the natural variability baseline. Moreover, this approach aggregates the statistics of one physical model, rather than the statistics and physics of different models. Therefore, it maintains the physical consistency of the projections, which is particularly relevant for nonlinear processes such as ice formation/melting and lake mixing, which also depend on high-frequency (e.g., diurnal) processes.

Given the challenges in deconvolving forced anthropogenic trends and natural variability[21], we choose to analyze output from a Large Ensemble of greenhouse warming simulations[23] conducted with the Community Earth System Model version 2[24] (CESM2-LE), for which 100 members have been run over 1850–2100 under a historical/SSP3-7.0 pathway (see "Methods"). We use here a subset of 90 of the 100 ensemble members for which daily lake ice thickness was saved as an output variable. This facilitates a definition of the forced signal as the ensemble mean of daily mean output across the 90 members, facilitating identification of forced trends in the phasing of seasonality in addition to mean state changes. The forced response is determined as the 90-member ensemble mean, whereas the natural variability is characterized by the amplitude of the spread across the ensemble members. For our analysis, we exclude lakes from Antarctica and Greenland (see "Methods").

CESM2 shows high fidelity to the observed climatological lake ice phenology when compared with observational records with more than 20 years of duration. The observational records were extracted from the Global Lake and River Ice Phenology dataset[25] (see "Methods"). The spatial variance in terms of climatological mean ice duration, ice freeze date, and ice breakup date are well reproduced by the fully coupled lake model, and the correlation coefficients ($r$ value) are 0.94, 0.78, and 0.94, respectively (Supplementary Fig. S1). The corresponding mean absolute errors (MAE) between observations and simulations are about 17, 12, and 16 days (Supplementary Fig. S1). The modest bias in the simulations toward the shorter duration of winter ice cover can be attributed to the existing warm bias of surface air temperature in the fully coupled model (see "Methods"). Ice thickness is validated using 15 observational records from Canada[26] (see "Methods"). The observed annual maximum ice thickness is reproduced well by the simulation with a high spatial correlation coefficient ($r$ value = 0.93), even though the model tends to underestimate the amplitude of seasonal to interannual ice thickness variations (linear regression slope = 2.1, Supplementary Fig. S1f).

## Results

### Changes in ice phenology and ice thickness

The ice freeze date and breakup date for lakes with an ice duration of longer than 5 days are defined as the first and the last days of the lake ice cover, respectively. The simulated timeseries of lake-area weighted global mean ice phenology, including ice freeze date (Fig. 1a), ice breakup date (Fig. 1b), and duration (Fig. 1c), exhibit some multi-decadal variability and short-term positive anomalies in the ensemble mean, which can be partly attributed to both anthropogenic aerosol forcing and some volcanic eruptions, respectively. Starting in the 1970s greenhouse warming effects outpace the cooling effect of anthropogenic aerosols, leading to a steady downward trend in lake ice duration, early melting and late ice formation. On average, lakes will freeze $20 \pm 8$ days later (uncertainty conveyed through the ensemble mean plus/minus two standard deviations of the ensembles, so are the same for the values afterward) in 2100 CE as relative to modern conditions. The global mean lake-area averaged ice breakup date is projected to advance by $20 \pm 7$ days over 2020–2100 CE. Both effects together lead to a shortening in global mean lake ice cover duration by $38 \pm 11$ days.

The trend pattern of simulated ice phenology in the 21st century reveals some important regional features (Fig. 1e–g). Poleward of the tropical/subtropical zone of no or only intermittent climatological 20th-century lake ice (lakes which have at least one ice-free winter out of 10 winters are indicated by red dots in Fig. 1g), we find an area in which lakes will switch from climatological winter coverage to intermittent coverage towards the end of the 21st century (Fig. 1g, cyan dots). This area accounts for $11.7 \pm 1.6\%$ of the total surface area of global lakes. The projected shortening in lake ice duration is particularly strong over the Tibetan Plateau and the Canadian Arctic with values exceeding $-0.45$ days per year. Over Norway and Sweden, the reduction of lake ice is less pronounced, attaining values of only $-0.28$ days per year. In the Southern Hemisphere (excluding Antarctica), lake ice coverage only occurs at the southern tip of the Andes (inset in Fig. 1g). However, 20th-century climatological ice cover in this area is very thin and its duration is very short. Towards the end of the 21st century, almost all the lakes in this region will have only occasional lake ice, and some are projected to lose their ice cover permanently.

As seasonal ice duration will shorten in response to greenhouse warming, lake ice mass will also decrease rapidly, as illustrated by the timeseries of annual maximum ice thickness (Fig. 1d). Maximum ice thickness remains relatively unaltered over 1850–1980. Subsequently, we see a marked negative trend that leaves the envelope of simulated ensemble natural variability by 2000–2010. The model projects that approximately $0.23 \pm 0.07$ m of global mean maximum lake ice thickness will be lost over the next 80 years. The spatial characteristics of ice thickness changes (Fig. 1h) are similar to those for ice duration (Fig. 1g) with the strongest trends of $-0.04$ m per decade occurring over the Canadian Arctic, along the northern Siberian and the Barents Sea Coast and on the Tibetan Plateau. Thinning of ice will increase the risk of potential accidents associated with human outdoor activities and winter road networks on lakes[27–29].

### Regional acceleration of lake ice loss and related mechanisms

Anomalies of ice duration and annual mean surface air temperature are highly correlated ($r$ value = $-0.8$, $P$ value < 0.001) (Fig. 2a; for more information, refer to Supplementary Note 1 and Fig. S2). This correlation also exists in the observations ($r$ value = $-0.47$) (Fig. 2b). On average, in the observations, ice duration decreases by 8.9 days for a 1 °C local warming, and in the model, we find a sensitivity of 9.9 days for the same period as in observations. The 1-day deviation may be related to a variety of factors, such as biases in the melting processes within the ice module[20], or the lack of lake surface morphology changes[6]. Overall, however, the sensitivity of lake ice to temperature on a global-scale is captured quite well in the model, as further

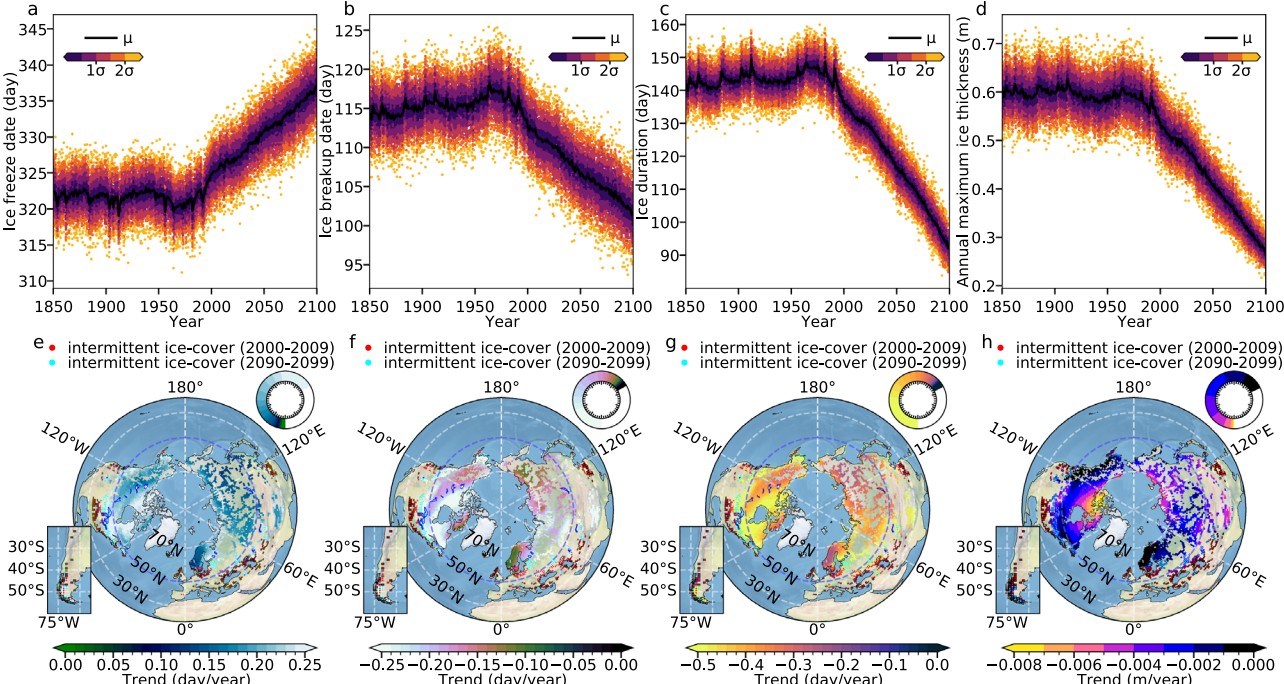

**Fig. 1 | Projections of lake ice phenology and thickness.** Timeseries of lake-area weighted global mean ice freeze date (**a**), ice breakup date (**b**), duration of ice cover (**c**), and maximum ice thickness (**d**) in each year in the CESM2-LE simulations. Black lines represent the 90-member ensemble mean (μ). Shaded bands indicate the standard deviations (σ) across ensemble members (circles). The y axes in (**a**) and (**b**) represent the day of the year. **e**–**g** are the patterns of the temporal trends of the ensemble mean for ice freeze date, ice breakup date, and ice duration during 2000–2100, respectively. Intermittent frozen lakes (defined as at least one winter without ice within 10 years) are represented by red points (2000–2009) and cyan points (2090–2099). Lakes without ice coverage for the period 2000–2009 and lakes in Antarctica and Greenland are not shown in the plot. **h** Trends of the ensemble mean of maximum lake ice thickness over 2000–2100. Pie charts in (**e**), (**f**), (**g**), and (**h**) show the spatial percentage of lake surface area corresponding to the values indicated by the color bar. The total extent of the pie charts represents the global lake area (excluding the Antarctic and the Greenland lakes).

indicated by the inset of Fig. 2b. Our global-scale analysis of the sensitivity of ice phenology to surface warming is also qualitatively consistent with previous studies[4,7,30–33]. Whereas natural variability has profound impacts on local ice phenology[34–41], our global-scale analysis further highlights the impacts of human-induced warming on lake ice loss and the associated thinning of lake ice cover (r value = −0.72, Fig. 2c), as indicated by the scatter diagrams (Fig. 2a, c).

By comparing the trend pattern for air temperature change (Fig. 2d) with the rate of ice loss (Fig. 1g, h), we find robust spatial correspondence. The areas exposed to the largest warming trends also experience the most rapid loss of lake ice (the cyan dots denoting lakes with loss rates over 0.45 days per year in Fig. 2d). Some areas in the Canadian Arctic are projected to warm with a rate of 0.1 °C per year, yielding ~8 °C warming anomaly by 2100. In these rapid ice loss areas, surface air temperature change explains over 70% of the ice duration variance (Supplementary Fig. S3). In contrast, ice loss in Scandinavia and Iceland is slow, with annual rates of −0.2 days per year in ice cover duration (Fig. 1g) and −1 mm per year for maximum ice thickness (Fig. 1h). The slow retreat of lake ice is related to the moderate surface air temperature change (0.02 °C per year or lower) in the eastern North Atlantic, related to the expansive "warming hole" due to the slowdown of Atlantic Meridional Overturning Circulation (AMOC) in the CESM2-LE[23] (Supplementary Fig. S5).

To better understand the spatial heterogeneity of lake ice trends, we focus on the seasonal evolution of lake ice indicators and their drivers. In the Canadian Arctic, northern Siberia and the Barents Sea surroundings, polar amplification is mostly a cold-season phenomenon[42], which can be related to the projected sea-ice loss in the Arctic Ocean and Hudson Bay. Model simulations and reanalysis data reveal that the enhanced polar warming over sea-ice areas in cold season can be explained by the so-called "heat capacitor" feedback[42]:

the opening of sea ice in summertime leads to the absorption of anomalous heat by the ocean. In wintertime, the excess heat can reduce sea-ice coverage (Fig. 3, first row) which in turn generates large heat fluxes from the warmer ocean to the highly stratified winter atmosphere. The atmospheric heating spreads further to neighboring land areas, where it can influence lake ice, peaking about three months later (Fig. 3, second row and Fig. 4a), and in accordance with the temperature/lake ice sensitivity relationship (Fig. 2a, c). This effect is most pronounced over the Canadian Arctic, parts of northern Siberia, and the Barents Sea surroundings. Our analysis reveals a clear link between Arctic Ocean sea-ice loss and Arctic land lake ice loss (Fig. 3).

Another important feedback for lake ice loss south of the Arctic Circle is the ice-albedo feedback. Due to future ice loss and reduced surface albedo, lakes will absorb more shortwave radiation (Fig. 3, fourth column) in the extended ice-free season. This effect is particularly strong over the Tibetan Plateau, where lake ice loss and strong mean insolation in autumn lead to a delayed onset of ice growth and absorption of anomalous heat by the lake water body (Fig. 3, fourth row). The excess heat can reduce ice cover in winter, therefore leading to the advancement of ice melting, which in turn triggers a positive feedback in spring. In addition, although future changes of lake ice albedo in springtime in the Tibetan Plateau are similar to those in the Canadian Arctic, the excess of absorbed solar radiation due to ice loss is much larger over the Tibetan Plateau (~20–30 W/m² in spring and 14 W/m² in the annual mean) in comparison with northern Canada (Fig. 4a, b). The reason can be attributed to the fact that the climatological mean solar radiation in spring is weak in the high latitudes, but much stronger over the Tibetan Plateau. The overall processes, including both "heat capacitor" feedback and "ice-albedo" feedback, are summarized schematically in Fig. 4c.

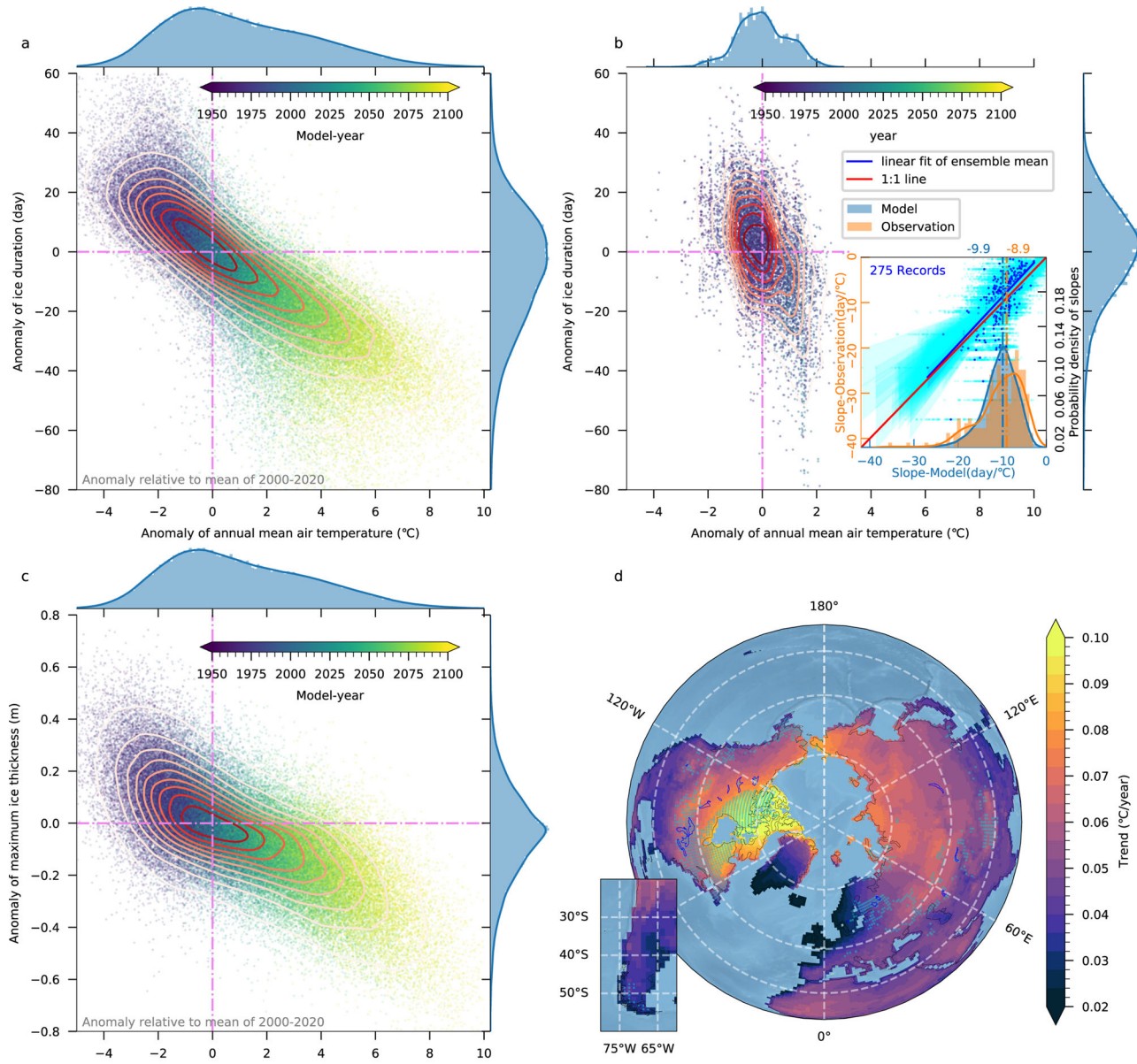

**Fig. 2 | Forced versus natural variability in lake ice. a** Scatter plot of annual mean air temperature anomaly against ice duration anomaly for individual lakes over the period 1950–2100 (anomalies are calculated relative to the 2000–2020 reference period). The simulation year is represented by dot color. A sample of 100,000 data points presented in this chart is chosen randomly from the total sample of 42,525,000 data points of ice-covered lakes and 90 ensemble members. The marginal plots on the right and top sides are normalized histograms of ice duration and annual mean air temperature, respectively. Contour lines covering scattered points represent a kernel density estimate of the joint probability density distribution of ice duration and air temperature. The darker the contour, the higher the probability density. **b** Same scatter plot as (**a**), except for the observed ice duration data obtained from the Global Lake and River Ice Phenology dataset[25] and surface air temperature taken from the CRUTEM4 dataset[69]. All lake ice observations used here cover at least 20 years since 1950. The inserted scatter plot shows the temporal

least-squares regression slopes (*P* value <0.1) of individual lakes between the annual mean surface air temperature and ice duration. The *y* axis is the slope in the observations (275 records) and the *x* axis is the slope in the model for the nearby lakes and same period. Cyan points represent value of one ensemble member, and solid blue points represent the ensemble mean. Lines and shaded bands represent the regression line and 95% confidence interval of the least-squared regression between the observation and the model. Blue line indicates the linear regression of the ensemble mean. The blue histogram represents the distribution of slopes in the model. The orange histogram is for the observations. The average of slopes is denoted by the vertical dashed line. **c** Same scatter plot as (**a**), but the *y* axis is replaced by the anomaly of the annual maximum ice thickness. **d** The linear trends of ensemble means of annual mean land surface air temperature during 2000–2100. The cyan points denote lakes with ice duration loss trend over 0.45 days per year.

## The emergence of no-analog lake ice phenology conditions

Future lake ice loss will ultimately impact lake biodiversity[17,43] by expanding pelagic habitats and extending the growing season for warm-water species, while increasing environmental stress and selection pressure for cold-water species that might lose their habitats and their competitive advantage over other species. To further quantify the projected loss of ice, we compare the projected lake ice duration

against natural variability, which is estimated in our case from the 90 ensemble members and the temporal variability between 1850 and 1950. In each lake, we consider the 2-standard deviation (σ) range around the temporal and ensemble mean value (μ) of ice duration variability during the 1850–1950 period (Fig. 5a) as a representation of the natural range that aquatic species have adapted to. In most lakes the ice duration starts to deviate for the first time from its natural range

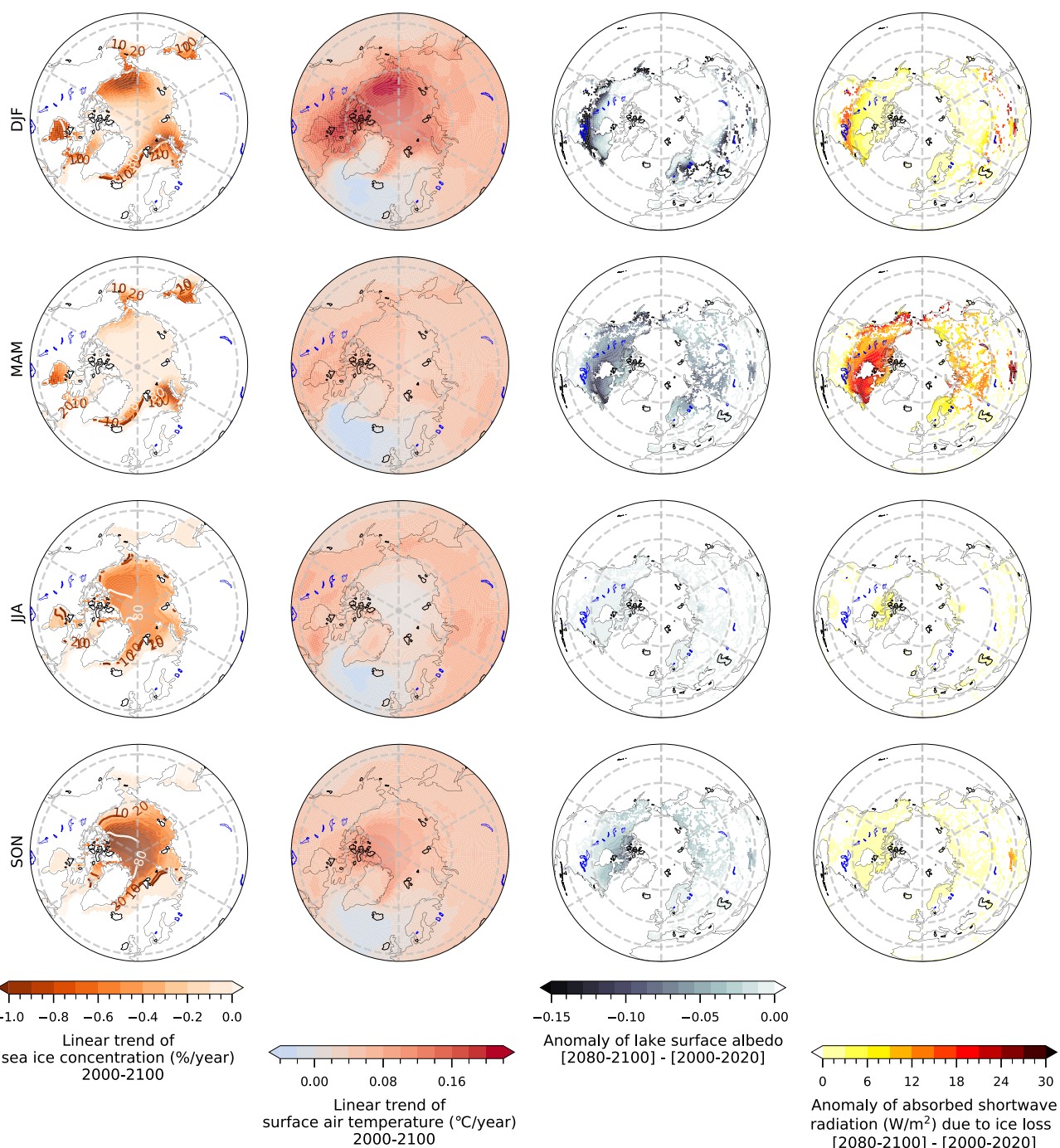

**Fig. 3 | Drivers of lake ice loss.** Seasonal linear trend of Arctic sea-ice concentration (first left column, unit: %/year) and surface air temperature changes (second column, unit: °C/year) in 2000–2100. Changes (climatological mean of 2080–2100 minus climatological mean of 2000–2020) of lake surface albedo (third column) and absorbed shortwave radiation (fourth column, unit: W/m²) by the lake due to ice loss. All analyses are performed on the ensemble mean fields. DJF, MAM, JJA, and SON denote the seasonal means from December to February, March to May, June to August, and September to November, respectively. This analysis is based on the full set of 60 ensemble members for which appropriate output is available (see "Methods").

(the lower limit of natural range, i.e., μ − 2σ) between 1980–1990 (Fig. 5b). Even though ice duration begins to deviate from its natural range in the last quarter of the 20th century (Fig. 5b), there is a 95% possibility of returning to the natural range, given the normal distribution of internal variability represented by the spread among ensemble members.

If, however, the projected ice duration in each lake drops below the lower 2σ limit of the natural variability range, the probability of ice duration returning to the natural range is only 2.5% (Fig. 5a for illustration). In this case, the ice phenology will become a no-analog situation for the local aquatic ecosystems (relative to 90 ensemble members simulated by the CESM2-LE) (Fig. 5c). Aquatic species, especially cold-water species that are highly adapted to under-ice conditions, would have to adapt to or colonize colder refugia, and the niches will be occupied by warm tolerant species[19]. We also calculate the time between first threshold exceedance (Fig. 5b) until reaching the no-analog situation (Fig. 5c) as the ratio of internal variability to the anthropogenic-forced trend. This calculation is based on the underlying premise that aquatic ecosystems in lakes with larger

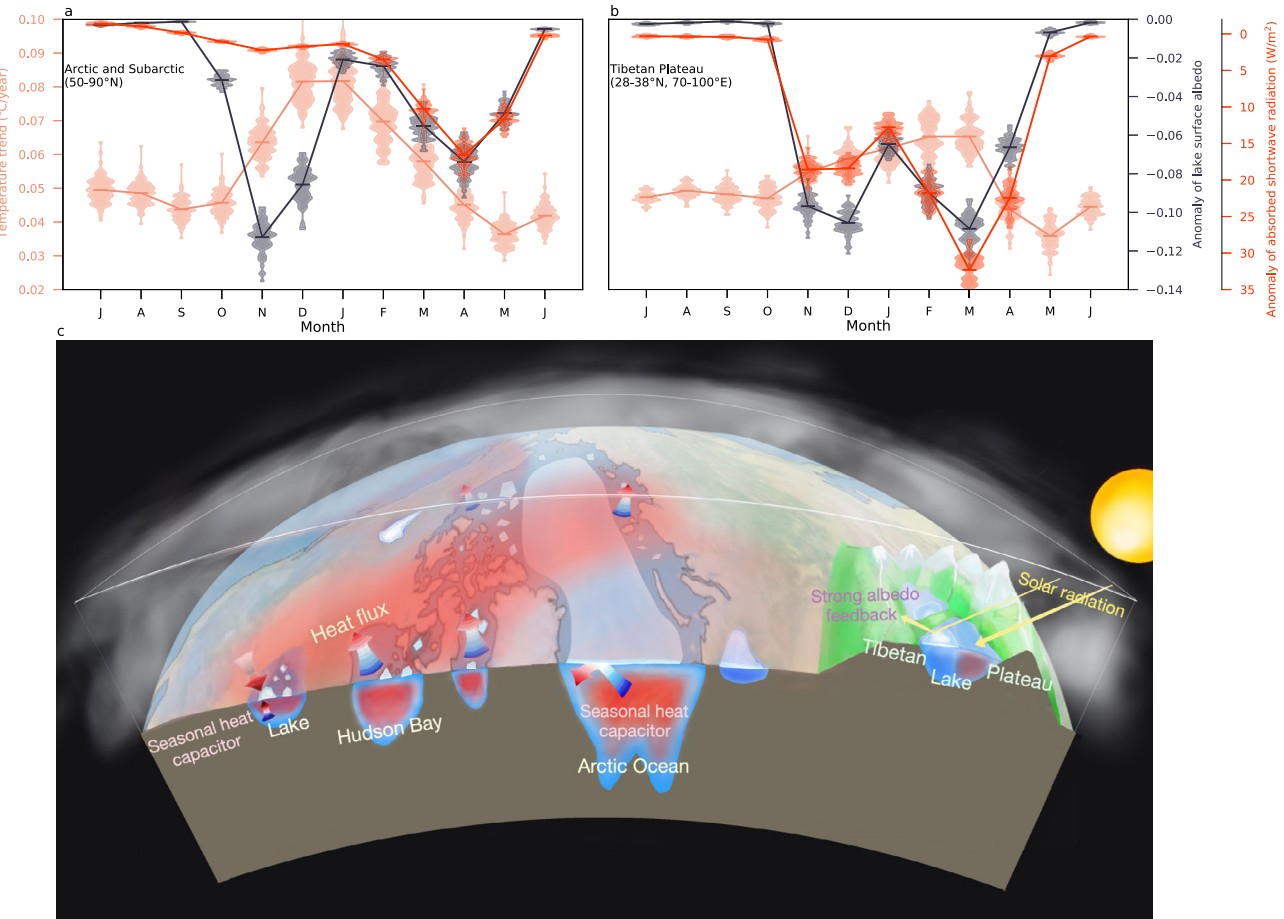

**Fig. 4 | Main drivers of projected lake ice changes.** Average of monthly linear trend of surface air temperature, anomaly of lake surface albedo, and anomaly of absorbed shortwave radiation due to ice loss in the Arctic and Subarctic (**a**) and in the Tibetan Plateau (**b**), respectively. The violin diagram illustrates the kernel probability density of ensemble members, and the solid line represents the ensemble mean. **c** Schematic of thermodynamic drivers of future lake ice decline. Blue-red arrows represent net surface heat fluxes. Red shading in the Arctic Ocean, Hudson Bay and in lakes indicates autumn and winter-time heat storage. Dark (light) red shading over land indicates intensified (weaker) warming due to polar amplification processes. Lakes in the Canadian Arctic are influenced strongly by sea-ice retreat and winter-time warming over the Arctic Ocean and the Hudson Bay. Rapid lake ice retreat over the Tibetan Plateau is amplified by a very large bi-seasonal lake ice-albedo feedback due to high incoming solar radiation. The background map is highly idealized.

natural variability have stronger resilience or greater tolerance to anthropogenic-induced climate changes. Species in lakes with narrow transition time windows, i.e., lakes with rapid lake ice loss and small natural variability, are likely to experience more intensive selection pressure and environmental stress. Lakes in the Canadian Arctic and western Siberia have a transition time of approximately 90 years and may experience the no-analog habitat near 2080 with respect to ice duration (Fig. 5c, d). Strikingly, lake ecosystems on the Tibetan Plateau have a transition time of only about 60 years and may be exposed to extreme habitat stress by the year 2040. The unprecedented shift in this region can be attributed again to the strong lake ice-albedo feedback on the Tibetan Plateau and the high incident subtropical solar radiation (Figs. 3 and 4). Other lakes outside these areas may not be subjected to the no-analog conditions within the 21st century, even though they will still experience substantial thermal stress. In Western North America, large natural variability (Supplementary Fig. S4) contributes to this later emergence, whereas the weaker signal-to-noise ratio over Scandinavia and Iceland is due to a combination of extensive natural variability and weak warming associated with the projected slowdown of the AMOC and thus weakening of northward heat transport in the CESM2-LE simulations (Supplementary Fig. S5).

## Discussion

A critical challenge in identifying forced changes in lake conditions is disentangling large natural variability for anthropogenic trends. To this end, we have invoked a 90-member subset of a Large Ensemble conducted with a state-of-the-art Earth system model. This has enabled us to identify the time of emergence of anthropogenic trends and the time at which no-analog conditions are reached. Two main hotspots where lake ice will be lost most rapidly and the underlying mechanistic drivers are identified are the Canadian Arctic and the Tibetan Plateau. Arctic and Hudson Bay sea-ice loss and the corresponding cold-season polar amplification of nearby land points accelerate lake ice decline of the Canadian Arctic (heat capacitor feedback). Over the Tibetan Plateau the most rapid ice loss occurs in spring and autumn when strong lake ice-albedo feedback associated with high climatological incoming solar radiation delays the formation of ice. This leads to more absorption of heat by the lakes, which further inhibits seasonal lake ice formation.

The seasonal cycle of temperature, mixing, light availability, and ice coverage can orchestrate important biological processes in lakes. Observations substantiate that below-ice ecosystems play a fundamental role in aquatic biodiversity[10,44,45]. The corresponding species-dependent phenology is oftentimes tied to the presence or absence of ice, which in turn impacts the seasonal cycle of prey and predators[43].

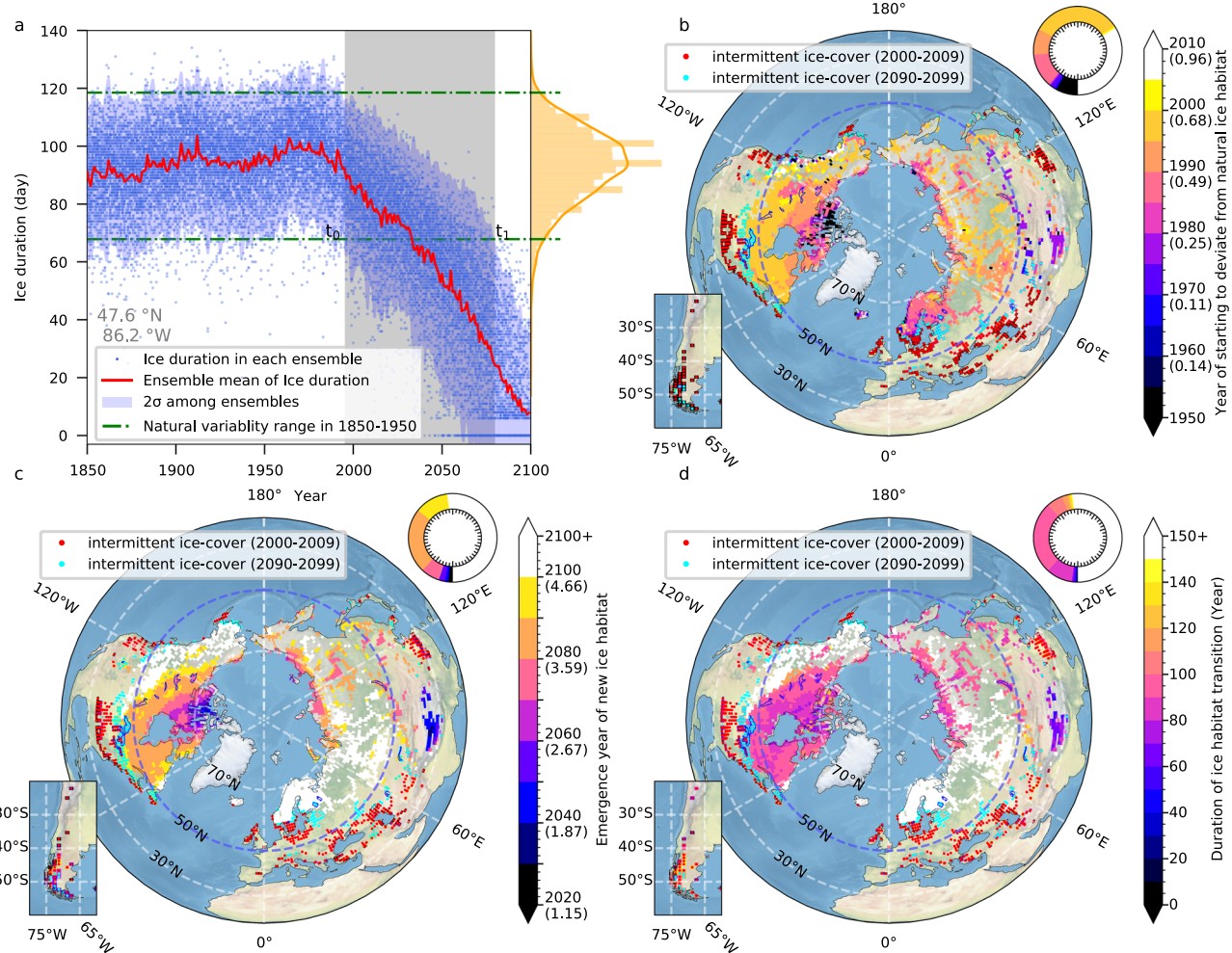

**Fig. 5 | Emergence of no-analog lake ice conditions. a** Timeseries of ice duration at a specific lake point (47.6°N, 86.2°W). Each blue dot represents the ice duration of individual ensemble members. The red solid line indicates the ensemble mean of ice duration. The orange histogram shows the probability distribution of the ice duration during 1850–1950. The two green horizontal lines represent two standard deviations around the mean. The difference between the two lines is interpreted as the natural habitat variability range. The left boundary of gray box represents the emergence year ($t_0$) of the anthropogenic signal when ice duration starts to deviate from the range of natural variability (defined as the first year when $\mu - 2\sigma$ (the ensemble mean minus two standard deviations) crossing the lower limit of the natural variability range, i.e., the lower green line, with at least five consecutive years). The right boundary of the gray box is the year ($t_1$) when ice duration completely deviates from the natural variability range (defined as the first year when

$\mu + 2\sigma$ crossing the lower limit of the natural variability range). The width of the gray box represents the maximum adaptation time ($t_1 - t_0$) for species to adapt to no-analog lake ice conditions in terms of ice duration. **b** Spatial distribution of the year when ice duration starts to deviate from the range of natural variability. Values in the brackets of the color scale are the global mean surface air temperature anomalies of the corresponding years relative to the climatological mean of 1850–1950. The relationship between time and global mean temperature anomaly is illustrated in Supplementary Fig. S6. **c** Spatial distribution of the year when ice duration completely deviates from the range of natural variability, that is, the emergence year of no-analog lake ice habitat. Values in the brackets are also the global mean surface air temperature anomalies as in (**b**). **d** Spatial distribution of transition time for species to adapt to the new habitat (or migrate poleward, if possible).

Ice-related phenology also influences nutrient cycling[46,47], dissolved oxygen[48], and the timing of algal blooms[49]. In many cases, the cascading effects of lake ice can impact ecological processes even in summertime[44]. In accordance with our projection of $38 \pm 11$ days reduction in ice duration by the end of this century, a recent study[50] projected a prolongation by $33 \pm 12$ days of lake stratification of the Northern Hemisphere under a similarly high emissions scenario (RCP 8.5).

Our CESM2-LE simulations show that the first anthropogenic deviation from the natural ice phenology range occurred already between 1980–1990 (Fig. 5b), after which lake ice loss rates are the fastest in terms of ice duration over the last century in both simulations (Supplementary Fig. S7b) and continuous observations from 30 lakes in the Northern Hemisphere[16]. In comparison, ice loss rates were insignificant in the first half of 20th century when the impact of anthropogenic forcings was minimal (Supplementary Fig. S7a).

The time when the trends become inconsistent with the natural ice phenology range corresponds to a breakpoint identified in phytoplankton assemblages in Arctic and Subarctic lakes[51–54]. Changes in phytoplankton assemblages have been attributed recently to the loss of lake ice in these regions[17,49,51,54–57], which led to extended growing seasons, habitat expansion, increased light and nutrient availability. Moreover, some of these observed recent changes in the aquatic biology of high Arctic lakes have been unprecedented with respect to the past 300 years[58], or even 200,000 years, when considering longer-term paleo reconstructions[59].

The projected rapid decline of lake ice over the next 80 years will adversely impact lake ecosystems. We demonstrate here that the lake ice in many regions has already started to move out of its natural envelope, with no-analog situations developing in some regions within the next 4–5 decades under the SSP3-7.0 emission pathway. Although our simulations have been conducted under one emission pathway,

they are of more general value for assessing the emergence of no-analog conditions against overall warming levels (relative to the climatological mean of 1850–1950, Fig. 5). Our results indicate no-analog conditions start to emerge under global warming of 1.9 °C (Fig. 5c). This work underscores the elevated risk of adverse ecological consequences under sustained high anthropogenic emissions over the 21st century. To avoid emergent and unprecedented changes in lake ice phenology, it is essential to keep global mean surface temperatures below 1.9 °C.

We believe this work is an important step forward in modeling lake ice and its response to greenhouse warming. Nevertheless, several caveats should be noted when interpretating our results. Our simulations use a gridded simulation method. Each 1° longitude–latitude grid cell has one representative lake whose mean depth is determined by the area-weighted average of depths of all lakes in the grid cell. However, individual lakes within the grid cell could have lake ice dynamics different from the representative lake. This is due to, for example, elevation gradient and depth[2]. Lake depth determines thermal inertia and mixing regimes of the water column, thereby impacting freezing and thawing processes of lake ice cover. Deeper lakes tend to take longer to freeze than shallower lakes under the same climate forcing. Moreover, the representative lakes were simulated with a one-dimensional lake model. Horizontal heterogeneity in water temperature and ice cover was not simulated. This is important for large lakes in which water circulation and ice cover mechanics in the horizontal direction impact lake ice phenology. Longer fetches in large lakes also lead to later freeze and earlier breakup by increasing wind speed, thus lacking horizontal features is likely to result in unrealistic ice phenology particularly in large lakes. However, the simulation of the representative lake does reflect changes in the most common lake type within the grid cell. Among all the factors like depth and lake size, air temperature is the dominant factor inducing lake ice changes according to a comprehensive analysis based on global observations[2], and these insufficiencies in one-dimensional models mainly impact the climatological mean more than variability of lake ice phenology. Long observation records from three lakes in one grid cell (1° × 1°) in Finland (Supplementary Fig. S8) show large differences in their climatological means of ice duration, but their long-term changes are similar and captured well by the representative lake simulation. Although the simulation tends to have larger bias in climatological mean ice duration in large lakes, such as Lake Baikal and Lake Superior (Supplementary Fig. S8), long-term changes, i.e., lake ice loss, over the last century are captured reasonably well by our simulation.

It should also be noted that lake ice growth is sensitive to the timing and depth of snow on ice surface[10], through the influence on ice thickness as well as altered albedo. While a 5-layer snow module is included in our model, uncertainty of snowfall derived from the atmospheric model could lead to inaccurate lake ice simulations. However, the full coupling between lakes and the atmosphere in our simulation does represent the most advanced modeling practice for lake ice dynamics to date by simulating the three-way interactions between water, ice and air and by resolving the diurnal cycle of lake thermodynamics. Importantly, our results allow us to separate natural variability in lake ice dynamics for the first time through the large ensemble simulation, which also adds confidence to the quality of our projections about responses of lake ice phenology to anthropogenic forcings. It is our hope that future work with single-forcing ensemble simulations[60] will enable further attribution of the respective roles of greenhouse gases and anthropogenic aerosols in modulating the response of lakes to anthropogenic forcing.

## Methods
### CESM2 large ensemble simulation model design
The ICCP/CESM Large Ensemble simulation[23] (CESM2-LE) is based on the Community Earth System Model[24] (CESM, version 2), with a nominally 1° × 1° degree horizontal resolution. The model is forced by the Coupled Model Intercomparison Project Phase 6 (CMIP6) historical forcing from 1850 to 2014 and the Shared Socioeconomic Pathways forcing scenario[61] (SSP3-7.0) from 2015 to 2100. The 90 members with daily lake ice thickness output fields (out of 100 ensemble members simulated) are initialized from a set of different combinations of initial states of ocean and atmosphere[23]. Among them, 80 members are initialized from 4 different years of a pre-industrial control simulation conducted with CESM2[24]: 1231, 1251, 1281, and 1301 (each initialization with 20 members) with micro-perturbations to the temperature field (i.e., combination of macro- and micro-initializations). In addition, 10 macro-perturbation members are initialized from 10 different years: 1011, 1031, 1051, 1071, 1091, 1111, 1131, 1151, 1171, and 1191 (each initialization with one member). Attribution analysis shown in Fig. 3 and Fig. 4 is based on the output fields of the last 10 micro-initializations (11–20) of 1231, 1251, 1281, and 20 micro-initializations of 1301, in addition to all macro-initializations. The Community Land Model[15] (CLM, version 5) used in this study is the land module of the fully coupled CESM2. CLM5 uses a nested subgrid hierarchy (containing grid cell, land unit, columns, and plant functional types) to represent the heterogeneity of the land surface. Each grid cell can have up to five one-dimensional land units (glaciers, lakes, urban areas, vegetated regions, and crops) which are modeled separately, and there is no horizontal flux exchange among the land units.

### The CLM5 lake model
CLM5's Lake, Ice, Snow, and Sediment Simulator[20] (LISSS) is a one-dimensional thermodynamic model, which solves a vertical thermal diffusion equation. A lake consists of snow (up to five layers), the lake water body (ten layers, consisting of liquid water and ice), soil (ten layers), and bedrock (five layers) in the vertical profile. Water volume of the lake water layer is fixed, and the thickness of the lake water layer is set according to the lake depth by the nominal method. Each grid cell has one lake modeled. The mean lake depth and lake-area fraction of the grid cell are aggregated from the high-resolution (0.05° × 0.05°) Global Lake and Wetland Database[62] and a global gridded lake depth dataset[63] to the CLM5 grid (0.9° × 1.25°). LISSS uses a bulk transfer approach to calculate the surface fluxes (momentum fluxes and heat fluxes) between the lake surface and the atmospheric boundary layer. The mixing processes within the lake water column include wind stirring, convection, molecular diffusion, and unresolved mixing processes. LISSS calculates the ice fraction in each layer of the lake body, instead of simulating a separate ice layer above water layers. To account for the effects of puddling and disintegration on the ice albedo, LISSS decreases the albedo of lake ice during the melting process. The albedo of snow is calculated by the method applied for non-vegetated land units. As a fully coupled lake model, the default timestep is 1800 s, so the diurnal cycle is resolved in our lake model which, however, is less represented in the offline-lake model simulations due to the temporal resolution of forcing fields. In our calculation, an ice-covered year is determined as a year in which ice cover lasts for more than 5 days. Lakes from Antarctica and Greenland are excluded in our analysis, as lake simulations there are subject to large uncertainties due to ice sheet processes in the two regions.

### Validation of ice phenology
The Global Lake and River ice dataset[25] contains 471 in situ observational ice phenological records for lakes in the Northern Hemisphere, of which 331 records have been observed for more than 20 years since 1850 (Supplementary Data 1). These records (≥20 years) which have lake gridpoints nearby in our simulation are selected to validate the simulation results. For each record, we select in our simulations the same years as in the observations, even though the phase of modes of natural climate variability in the model is unrelated to that in the observations for any individual ensemble member. Global patterns of

the mean climatological ice freeze date, ice breakup date and ice duration are further validated (Supplementary Fig. S1). In the observations, the freeze date is defined as the first day when the lake is completely covered by ice. The breakup date is defined as the date of the last breakup observed before ice-free conditions. The ice duration is defined as the number of days for which the lake is completely covered by ice. Based on these definitions, the ice duration is not exactly the same as the difference between the breakup date and the freeze date. Therefore, the MAE of ice duration can be different from the sum of the MAE of freeze date and breakup date.

In the simulations, ice cover is assumed to be homogeneous horizontally. The ice freeze date is determined as the first day of ice cover appearing (ice thickness > 0 m), the ice breakup date is determined as the last day of ice cover disappearing (ice thickness = 0 m), and the ice duration is determined as the total number of days with ice cover which is roughly equal to the difference between the breakup date and the freeze date. In reality, freeze dates and breakup dates in different parts of the lake might be different due to the heterogeneity of ice cover, but the in situ observations oftentimes represent ice phenology in the shoreline area covered by the observer's sight. Therefore, this heterogeneity may lead to the deviation between observations and simulations, especially for large lakes, as the larger lakes tend to have higher heterogeneity in ice cover. Notably, the long freezing process contributes to this bias. Consequently, as shown in Supplementary Fig. S1 the freeze date was relatively poorly (r value = 0.78) reproduced by the simulation. By comparison, the melting process is rapid, so the breakup date is more consistent (r value = 0.94) between observations and the CESM2-LE simulation.

Although the method of using one representative lake for each grid point may overlook the influence of lake-specific factors (such as morphology and discharge), the high correlation between observations and simulations implies such influences are small, because to a first order the spatial and temporal variance are controlled by climatic factors. Additionally, timeseries of 16 long records are validated, respectively (Supplementary Fig. S8). Please note that the fully coupled model can of course not reproduce the observed trajectory, but only its statistics. Supplementary Fig. S8 shows that the range of variability in these records is well reproduced by the CESM2-LE simulations. There are three long observational records nearby the grid point of [61.7°N, 23.8°E] (the last panel in Supplementary Fig. S8). Although each lake has its own characteristics, the long-term changes of these three records are well captured by the simulation at this grid point.

Supplementary Fig. S1 reveals that the lake model in CESM2 somewhat underestimates the duration of lake ice cover. In the CESM2-LE simulations, a large part of the difference can be attributed to a warm bias in the simulated land surface air temperature. When compared with the land surface air temperature in the ERA5 reanalysis dataset[64], the simulated air temperature of CESM2 during 1981–2020 is 1.5 °C higher in the 30°N–70°N band where most of the lakes studied are located. As shown in Fig. 2b, when the annual mean air temperature rises by 1 °C, the ice duration decreases by 9.9 days. A warming bias of 1.5 °C therefore roughly translates to a bias of −14.9 days in ice duration. In addition, natural variability may play an additional role in the offset between model and observations.

## Validation of ice thickness

Ice thickness in the simulations was validated by in situ observational records collected as part of the Canadian Ice Thickness Program during 1947–2002[26]. Ice thickness was measured using hand drillers at the same location every year. Annual maximum ice thickness is obtained from weekly measurements during ice-covered periods (e.g., Supplementary Fig. S1e). In total, 15 records have weekly measurements (Supplementary Data 2) with annual maximum thickness ranging from

0.4 to 2.2 m. The climatological mean of the annual maximum ice thickness was compared with the nearby grids in our simulations, and again we choose the same periods for simulations and observations. The spatial variance of the annual maximum ice thickness is reproduced well by our simulation as indicated by a high pattern correlation coefficient (r value = 0.93; Supplementary Fig. S1f). Nevertheless, our lake model tends to underestimate the variance toward thicker ice cover, with the regression slope of 2.1 (Supplementary Fig. S1f). In addition to the existing warm bias in the fully coupled model, other more specific lake model biases could contribute to this discrepancy.

## Performance comparison with the offline-lake model

As a comparison, we also evaluate the lake ice simulation conducted by the SimStrat-UoG[65] (one-dimension, offline) model in the Inter-Sectoral Impact Model Intercomparison Project phase 2b[66] (ISIMIP 2b) using the Global Lake and River ice dataset. In the ISIMIP 2b, there are five lake models in total, and SimStrat-UoG is the best model regarding its performance on ice phenology[7] (Supplementary Fig. 27 in ref. 7). This project used a grid-simulation method similar to CLM5, and used the same bathymetry database[63]. However, the spatial resolution there is 0.5° × 0.5° relative to 0.9° × 1.25° used for CLM5. The SimStrat-UoG model is forced by the daily output fields, including surface air temperature, surface wind speed, longwave and shortwave radiation, and specific humidity from GFDL-ESM2M, HadGEM2-ES, IPSL-CM5A-LR, and MIROC5 from the CMIP5 project. All the fields were interpolated to a 0.5° × 0.5° grid and then were bias-corrected against the ERA-Interim reanalysis data. The MAE for the climatological mean ice duration is 15.7 days (Supplementary Fig. S9), which is close to the value of our simulations (17.3 days). As is discussed above, most of the bias in our simulations can be attributed to the warm bias existing in the fully coupled model. Lake temperatures in the CESM2-LE are not bias-corrected, in contrast to the procedure employed with the offline SimStrat-UoG model simulation. Although the spatial resolution of the SimStrat-UoG simulation is twice that of CLM5, the bias caused by its model deficiency is even higher. Clearly, when compared with SimStrat-UoG model (r value = 0.77; Supplementary Fig. S9), CLM5 has better prediction skill (r value = 0.93; Supplementary Fig. S1) in ice thickness variance, although the SimStrat-UoG model does not have the trend of underestimation towards thick ice cover.

## Statistical methods

In the large ensemble simulations analyzed here, natural variability is represented by the standard deviation across the ensemble members, and the ensemble mean represents the forced response to the anthropogenic emissions. The changes in the ice duration, freeze date, breakup date, and annual maximum ice thickness over 2020–2100 are determined as the difference between the values of all ensemble members in 2100 and the values in 2020. The difference in the mean across all ensemble members is the forced response, and the standard deviation represents the natural variability, which itself can be influenced by greenhouse warming. Similarly, changes in intermittent ice coverage between 2090–2100 and 2000–2010 were first calculated for individual ensemble members. Then, the forced response is determined as the average changes over all ensemble members. Changes in lake surface albedo and absorbed shortwave radiation in Fig. 3 were determined in the same way. All the linear trends in Figs. 1–3 are based on 90-member ensemble mean fields. The correlation coefficients, MAE, slopes, and intercepts in our model/data validation are all based on the ensemble mean fields of our CESM2-LE simulations and the SimStrat-UoG model simulation (Supplementary Fig. S9). The correlation analysis is carried out by using the linear least-squares regression tool in the "Scipy" package[67]. The calculation of the time of emergence for the anthropogenic signal and no-analog situation in ice phenology is described in the caption of Fig. 5.

## Data availability

Source data are provided with this paper. The model output of CESM2-LE is available via: https://www.cesm.ucar.edu/projects/community-projects/LENS2/data-sets.html. The Global Lake and River Ice Phenology Database was archived from https://nsidc.org/data/lake_river_ice/. The Canadian Ice Thickness Program Data are available at https://www.canada.ca/en/environment-climate-change/services/ice-forecasts-observations/latest-conditions/archive-overview/thickness-data.html. The EAR5 reanalysis dataset was achieved from the Copernicus Climate Change Service's Climate Data Store (CDS; https://cds.climate.copernicus.eu/cdsapp#!/search?type=dataset). The CRUTEM4 dataset is available from https://crudata.uea.ac.uk/cru/data/temperature/. The ISIMIP 2b SimStrat-UoG simulation results were achieved from the Earth System Grid Federation (ESGF; https://esgf-data.dkrz.de/).

## Code availability

The code used to calculate ice phenology and generate the figures in this study is available at https://github.com/geohuanglei/CESM2-LE-lake-ice-projection[68].

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

## Acknowledgements

The CESM2-LE simulations were conducted on the IBS/ICCP supercomputer "*Aleph*" 1.43-petaflop high-performance Cray XC50-LC Skylake computing system with 18,720 processor cores, 9.59-petabyte storage, and 43-petabyte tape archive space. The CESM2 Large Ensemble (CESM2-LE) simulations presented here have been conducted through a partnership between the Institute for Basic Sciences (IBS) Center for Climate Physics (ICCP) in South Korea and the Community Earth System Model (CESM) Project at the National Center for Atmospheric Research (NCAR) in the US, representing a broad collaborative effort between scientists from both centers. L.H., A.T., S.-S.L., K.R., and R.Y. were supported by the Institute for Basic Science (project code IBS-R028-D1). E.-S.C. was supported by the Korea Polar Research Institute (KOPRI) grant funded by the Ministry of Oceans and Fisheries (KOPRI PE22010).

## Author contributions

L.H. and A.T. designed the study. L.H. wrote the initial manuscript draft and produced all figures. L.H., A.T., S.-S.L., K.R., R.Y., and E.-S.C. contributed to the interpretation of the results and to the improvement of the manuscript.

## Competing interests

The authors declare no competing interests.
