## [Peer Review File · Nature Communications]

Emerging unprecedented lake ice loss in climate change projectionsREVIEWER COMMENTS

Reviewer #1 (Remarks to the Author):

The authors detail the use of ensemble simulations with CESM and its contained LISSS lake model to simulate lake ice in response to atmospheric conditions and specifically the response to anthropogenic drivers. The primary result is that lake ice is projected to decrease over the next 80 years in both measures of ice duration and thickness. The most interesting result presented here is the break point for which lake ice (duration) moves out of the “natural habitat variability range” and into a “no-analogue” regime. Secondly, they report on two geographic regions where change is accelerated (Canadian Arctic and Tibetan Plateau). While this approach uses ensembles of coupled atmosphere/climate and lake (land surface) model, my primary concern is the fundamental results are similar to what’s been reported in recent literature (Sharma et al., 2020 GRL; and others) in regard to projected lake ice loss. An attempt is made to perhaps distinguish the coupled modeling approach from some other 1D modeling studies by comparing the results to an offline simulation of SimStrat driven by common climate forcings. While there is demonstration of improvement over the SimStrat simulation, it’s unclear if and how the results are fundamentally or significantly different in terms of ice loss. I think the authors could improve the manuscript by highlighting the novelty in this work and how specifically how it advances the field.

- Recent work by Sharma et al., (2020 Geophysical Research Letters), and others, have projected the loss of lake ice over the next century. What is this work during that is fundamentally different?

- Why are 90 of the 100 ensemble members used? What happened to the missing 10?

- Why motivates the cutoff threshold for an ice covered year (e.g., 5 days)? Does this threshold have any noticeable impact on the results?

- Page 11: In the observations, the freeze date is the first day when the lake is completely covered by ice (and ice duration is number of days completely covered by ice). Many lakes rarely see complete (100%) lake ice coverage, so by this measure would they be considered ice free even if at 99% coverage? It’s unclear if the simulation is being guided by the same definition for validation. And if it’s not, why not?

- Line 321: It appears ice thickness is being used for the model metric as to determine the presence of ice. Which means even very small value of ice thickness at even a small fraction (concentration) will designate the day as “ice covered”. First, this approach seems starkly different than how the observations are reporting freeze date and ice duration. Second, this requires the model to be accurate at these low ranges of thickness and concentration. This can be problematic for water (and air) temperatures marginally above or below freezing, or sensitive to cloud cover (shortwave radiation), or problematic when wave-ice interaction processes are not resolved, which I don’t believe LISSS includes. Furthermore, inaccuracies in bathymetric data (which global data sets certainly include) can influence thermal structure and ice growth. Ultimately, how sensitive are the results to this >0 ice thickness threshold? Has the CLM lake model (LISSS) been demonstrated to perform well at these values?

- Line 328: Are large lakes being considered here as part of the results or is there a cut off? It is well known that 1D thermodynamic lake models can perform poorly over large lakes due to the complexity of the physical processes but also due to the heterogeneity of ice cover.

Reviewer #2 (Remarks to the Author):

Overview

This is a reasonably good article which gives a fairly detailed insight into the main predictions for general trends in lake ice cover and thickness from an already published model. A series (~90) of model runs have been conducted with the same model. The authors argue that the spread of these runs captures natural variability and that the mean represents anthropogenic forcing, yet the mean could just be the mean of natural variability. Whilst this keeps consistency in the physical parameters within the model it obviously relies on just one model capturing physical processes, which could be a questionable approach. However, there is good validation of the model from Canadian lake data, although it should also be noted that lake temperatures and ice phenology response to climate can be heterogenous. There is very limited discussion of parameterisation of key factors that may affect lake ice phenology; such as changes in snowfall (potentially increasing ice thickness etc) and wind patterns/speeds. How does cloud cover affect radiative cooling of lake water and ice formation?

Key Results and Significance

There is some statistical analysis of trends in lake ice since the 1980s. Some more analysis and direct comparison to other periods of warming earlier in the 20th Century (i.e. how does recent lake ice loss compare to Early Twentieth Century Warming period?) would add substantial weight to the analysis of anthropogenic forcing and identification of emergent trends. Furthermore, direct comparison and discussion of model runs on an SSP1-1.9 with SSP3-7.0 would allow a clearer identification of anthropogenic forcing on future lake ice trends. Currently a more fitting title would be 'Analysis of Lake Ice Trends under High Emissions Climate Scenario (SSP3-7.0)' as analysis of past anthropogenic forcing on lake ice changes is limited (see comments) and comparison to model runs from a low emissions scenario in future would allow further clear identification of anthropogenic forcing.

Beyond this there are some substantial findings, particularly regarding predictions of lake ice loss in the Canadian Arctic and Tibetan plateau. The former is logically argued to be associated with loss of sea ice in the Arctic and consequent Arctic amplification of climate change during the 'cold season', which in reality will be particularly important during the Autumn (please be more specific than 'the Fall'). Given this important result and identification of feedback mechanisms, it is also evident from the figures that areas in the Russian Arctic surrounding the Barents sea sector have experienced lake ice loss too, which should be noted in the text. The identification of areas such as the Tibetan plateau that are predicted to have substantial further lake ice loss, and experience non-analogous ice coverage within the next 4-5 decades are important results, for ecology and also indigenous populations that may travel over them during winter.

Validity, Data and Methodology

The data and methodology are of a fairly high standard, particularly with the validation data. However, just using one model could be questioned. Although there is some comparison to other models. I am not a modeller so other reviewers comments on this should be favoured.

Analytical Approach

Analytical approach seems to be fairly sound. Although again I am not a modeller so cannot comment fully on this.

Suggested Improvements

There should be more of an introduction to the main factors affecting lake ice loss and an overview of how/which of these are parameterised in the model, which should be accessible to non-specialists. Several sections of the text should be reviewed so that the meaning is clearer for non-modeller scientists. Some restructuring to help build the argument in stages for the reader and not get lost in the modelling detail would substantially improve the accessibility of the article.

Clarity and Context

Generally reasonably good. The language needs to be changed in some sections, for results regarding predictions of future outcomes there is uncertainty, so 'will' should be changed to 'may'. Please avoid referring to 'the fall' and refer to seasons (autumn) and ideally specific months.

REVIEWER COMMENTS

Reviewer #1 (Remarks to the Author):

The authors detail the use of ensemble simulations with CESM and its contained LISSS lake model to simulate lake ice in response to atmospheric conditions and specifically the response to anthropogenic drivers. The primary result is that lake ice is projected to decrease over the next 80 years in both measures of ice duration and thickness. The most interesting result presented here is the break point for which lake ice (duration) moves out of the “natural habitat variability range” and into a “no-analogue” regime. Secondly, they report on two geographic regions where change is accelerated (Canadian Arctic and Tibetan Plateau). While this approach uses ensembles of coupled atmosphere/climate and lake (land surface) model, my primary concern is the fundamental results are similar to what’s been reported in recent literature (Sharma et al., 2020 GRL; and others) in regard to projected lake ice loss. An attempt is made to perhaps distinguish the coupled modeling approach from some other 1D modeling studies by comparing the results to an offline simulation of SimStrat driven by common climate forcings. While there is demonstration of improvement over the SimStrat simulation, it’s unclear if and how the results are fundamentally or significantly different in terms of ice loss. I think the authors could improve the manuscript by highlighting the novelty in this work and how specifically how it advances the field.

We thank the reviewer for their careful reading and constructive comments on our manuscript. We will address the comments one by one.

- Recent work by Sharma et al., (2020 Geophysical Research Letters), and others, have projected the loss of lake ice over the next century. What is this work during that is fundamentally different?

Previous lake ice projections were either based on statistical or physical-based offline lake models. Firstly, both statistical models and offline lake simulations cannot deconvolve forced anthropogenic trends and natural variability in lake ice. When no-analogue conditions of lake ice will emerge thus remains to be unanswered. Moreover, statistical models, e.g., *Sharma et al. (2019, 2021)*, cannot provide projection of changes in thickness and duration of ice cover. Although they can account for factors such as morphology, depth, and meteorological forcings, they usually ignore important feedbacks such as the ice-albedo and heat capacity feedbacks which are important in the formation of lake ice and thawing, respectively. Offline lake simulations (e.g., *Woolway et al., 2019, Grant et al., 2021*), forced by daily meteorological forcings, cannot resolve the diurnal cycle of lake thermodynamics and the three-way interaction between lake, lake ice and atmosphere.

We have added one paragraph in the introduction [Other studies that address lake ice responses to greenhouse warming were conducted with statistical models, which are built on the empirical relationship between the observational ice phenology and meteorological variables^{2,9}. Their statistical projections document that a large number of lakes are likely to experience intermittent or permanent winter ice cover loss in the Northern Hemisphere until the end of this century, but they do not provide projection of changes

in thickness and duration of ice cover. Physical process-based lake-ice model simulations, on the other hand, have been used in recent years^{7,8,10} to assess future climate change in lake systems. They run as offline models which are typically forced by daily meteorological data derived from climate model simulations. While thermodynamic processes of water and ice are resolved explicitly in these simulations, the three-way interaction between lake, lake ice and atmosphere, is not represented. Such coupling will play an important role in particular for large lake systems which are known to have considerable impacts on local climate¹¹⁻¹³. Another caveat in such offline model simulations is the fact that diurnal cycle thermodynamics are not resolved. Given these issues, the modelling community has made substantial efforts to include lake simulators into Earth system models as an interactive component^{14,15}.] to emphasize that our fully coupled large ensemble simulation is an advancement as compared to previous studies, in that it resolves the diurnal cycle of lake thermodynamics as well as three-way interactions between the air, ice and lake water. We also demonstrate that our CESM2-LE simulations outperform offline lake simulations with respect to observed climatologies and statistics (Material and Methods section).

More importantly, we have added one paragraph in the introduction [Lake ice has experienced unprecedented loss rates in the last decades¹⁶, and lake organisms are suffering severe changes in their habitat with shortening of ice duration which had led to reorganization of aquatic ecosystems, for example, in circumpolar lakes¹⁷. Exceeding climate-related thresholds of lake ice phenology can even accelerate such biological processes¹⁸. Lake organisms may be exposed to no-analogue ice conditions in the future which are unlikely under natural climates. What's more, disappearance of extant ice conditions will increase the risk of local extirpation of populations for cold species¹⁹. It is thus fundamental to project time and the corresponding global warming levels for the emergence of no-analogue conditions for risk assessment and adaptation in the future.] to emphasize that our fully coupled large ensemble simulation is a major advancement as compared to previous studies, in that it projects the emergence of no-analogue ice conditions.

Our revised manuscript now provides important new information on the novelty of our study and the modeling approach (Lines 33-57).

Sharma, S. et al. Widespread loss of lake ice around the Northern Hemisphere in a warming world. *Nat Clim Change* 9, 227-231, doi:10.1038/s41558-018-0393-5 (2019).

Sharma, S. et al. Loss of Ice Cover, Shifting Phenology, and More Extreme Events in Northern Hemisphere Lakes. *Journal of Geophysical Research: Biogeosciences* 126, e2021JG006348, doi: 10.1029/2021JG006348 (2021).

Woolway, R. I. & Merchant, C. J. Worldwide alteration of lake mixing regimes in response to climate change. *Nat Geosci* 12, 271-276, doi:10.1038/s41561-019-0322-x (2019).

Grant, L. et al. Attribution of global lake systems change to anthropogenic forcing. *Nat Geosci*, doi:10.1038/s41561-021-00833-x (2021).

- Why are 90 of the 100 ensemble members used? What happened to the missing 10?

The first 10 members only saved monthly data (these were the default CMIP6 settings) which cannot be used for ice phenology calculations. Only after the first 10 ensembles members completed did we decide

to take the opportunity with the large ensemble to study lake ice phenology. For the remaining 90 ensemble members opted to save daily output fields, and 90 members adequately sample the natural variability in a way that is sufficient for our calculations. We therefore expect no discernible differences for the ice phenology projections if we had accounted for 100 members, instead of 90.

- Why motivates the cutoff threshold for an ice covered year (e.g., 5 days)? Does this threshold have any noticeable impact on the results?

No, this cutoff does not change the large-scale trends of ice phenology. We set up this threshold with the purpose of keeping the simulations close to the observations. In the one-dimensional model, the assumption of homogeneity in the horizontal direction allows thin ice cover to exist. However, in the real world, such thin ice cover does not exist because it is very fragile and cannot last long due to factors like destruction by wind. The cutoff may impact intermittent ice-covered lakes but it does not change our estimation on the global scale.

- Page 11: In the observations, the freeze date is the first day when the lake is completely covered by ice (and ice duration is number of days completely covered by ice). Many lakes rarely see complete (100%) lake ice coverage, so by this measure would they be considered ice free even if at 99% coverage? It's unclear if the simulation is being guided by the same definition for validation. And if it's not, why not?

“Days with complete ice cover” is a practical metric that is commonly used by ice-observers. There are undoubtedly errors in the observations, since an observer’s visual horizon can be limited – in particular in large lakes. Certainly, extrapolating ice-covered conditions from few observational points to a large lake system can lead to biases.

The simulation is guided by the same definition. The one-dimensional lake model, as is used in our simulation, assumes homogeneity in the horizontal direction, i.e., ice cover with thickness over 0 cm is defined as 100% ice cover. In reality, thin ice cover is unstable and impossible to form complete ice cover. So, we set up a 5 day threshold to account for the differences between observer lake ice cover assessments and the 1-d configuration used in our model simulation. Although errors exist in both observations and simulations, the validation shows high consistency between them (Fig. S1 and S8).

- Line 321: It appears ice thickness is being used for the model metric as to determine the presence of ice. Which means even very small value of ice thickness at even a small fraction (concentration) will designate the day as “ice covered”. First, this approach seems starkly different than how the observations are reporting freeze date and ice duration. Second, this requires the model to be accurate at these low ranges of thickness and concentration. This can be problematic for water (and air) temperatures marginally above or below freezing, or sensitive to cloud cover (shortwave radiation), or problematic when wave-ice interaction processes are not resolved, which I don’t believe LISSS includes. Furthermore, inaccuracies in bathymetric data (which global data sets certainly include) can influence thermal structure and ice growth. Ultimately, how sensitive are the results to this >0 ice

thickness threshold? Has the CLM lake model (LISSS) been demonstrated to perform well at these values?

We concur regarding the inconsistency between the model and the observations for the ice cover for marginal ice thicknesses. As is discussed above, in order to match the timescales over which the simplified model can represent real-world ice behavior, we apply a 5 day threshold in addition to a 0 cm threshold to define the ice cover duration.

Although it is impossible to remove this inconsistency with a one-dimensional lake model, errors induced by the inconsistency are relatively small as compared to the rates of change due to meteorological condition changes. As our model and the observations show, 1°C local warming leads to a reduction of 8-10 days in ice duration (Fig. 2b).

While the one-dimensional lake model cannot simulate the full suite of real-world mechanisms (e.g., wave-ice interaction) and heterogeneity in the horizontal direction of ice cover, our fully coupled lake simulation largely advances the current lake simulations by resolving the diurnal cycle (30 min temporal resolution) of lake thermodynamics and the three-way interaction between lake, lake ice and atmosphere. For example, lake ice freezing at night and melt in daytime, cloud cover changes on the sub-daily scale impacting shortwave radiation budgets of lakes are all included. These processes are important for ice cover with marginal thickness particularly in large lakes but they have not been resolved in global-scale lake simulations prior to our study.

In summary, we add a thorough discussion (Lines 295-323) in the manuscript regarding factors influencing lake ice, and further describe the robustness and limitations of our simulations. A more sophisticated and realistic simulation of lake ice, including local growth and horizontal mechanics of ice cover would certainly be desirable in the future, (e.g., three-dimensional lake-ice thermodynamical model (e.g., Sun et al., 2020)), but at the current stage this class of models globally with 100 ensemble members is computationally prohibitive.

Sun, L., Liang, X.-Z., & Xia, M. (2020). Developing the Coupled CWRP-FVCOM Modeling System to Understand and Predict Atmosphere-Watershed Interactions over the Great Lakes Region. *Journal of Advances in Modeling Earth Systems*.

- Line 328: Are large lakes being considered here as part of the results or is there a cut off? It is well known that 1D thermodynamic lake models can perform poorly over large lakes due to the complexity of the physical processes but also due to the heterogeneity of ice cover.

In fact we do not cut off any lakes in our analysis. We also include big lakes (e.g., Lake Baikal and Lake Superior) in our validation as well (Fig. S1 and S8). Although the model tends to overestimate ice duration in the large lakes, the long-term changes in the last century are captured well by our 1-dimensional simulation. These lakes are composed of several grid cells (0.9°x1.25°) for our simulations, thus the horizontal heterogeneity is partly presented by our simulation albeit lacking horizontal circulation and advection.

Reviewer #2 (Remarks to the Author):

Overview

This is a reasonably good article which gives a fairly detailed insight into the main predictions for general trends in lake ice cover and thickness from an already published model. A series (~90) of model runs have been conducted with the same model. The authors argue that the spread of these runs captures natural variability and that the mean represents anthropogenic forcing, yet the mean could just be the mean of natural variability. Whilst this keeps consistency in the physical parameters within the model it obviously relies on just one model capturing physical processes, which could be a questionable approach. However, there is good validation of the model from Canadian lake data, although it should also be noted that lake temperatures and ice phenology response to climate can be heterogenous. There is very limited discussion of parameterisation of key factors that may affect lake ice phenology; such as changes in snowfall (potentially increasing ice thickness etc) and wind patterns/speeds. How does cloud cover affect radiative cooling of lake water and ice formation?

Thank you very much for the constructive comments.

Although our results only rely on one lake model, we also compare it to another model (SimStrat-UoG) which performed the most realistically in the ISIMIP-2b project (Grant et al., 2021). The validation shows our fully coupled lake model have comparable or even better performance than the SimStrat-UoG lake model [As a comparison, we also evaluate the lake ice simulation conducted by the SimStrat-UoG⁶⁵ (one-dimension, offline) model in the Inter-Sectoral Impact Model Intercomparison Project phase 2b⁶⁶ (ISIMIP 2b) using the Global Lake and River ice dataset. In the ISIMIP 2b, there are five lake models in total, and SimStrat-UoG is the best model regarding its performance on ice phenology⁷ (Supplementary Fig. 27 in Grant et al., 2021). This project used a grid simulation method similar to CLM5, and used the same bathymetry database⁶³. However, the spatial resolution there is 0.5°x0.5° relative to 0.9°x1.25° used for CLM5. The SimStrat-UoG model is forced by the daily output fields including surface air temperature, surface wind speed, longwave and shortwave radiation, and specific humidity from GFDL-ESM2M, HadGEM2-ES, IPSL-CM5A-LR, and MIROC5 from the CMIP5 project. All the fields were interpolated to a 0.5°x0.5° grid and then were bias-corrected against the ERA-Interim reanalysis data. The MAE for the climatological mean ice duration is 15.7 days (Fig. S9), which is close to the value of our simulations (17.3 days). As is discussed above, most of the bias in our simulations can be attributed to the warm bias existing in the fully coupled model. Lake temperatures in the CESM2-LE are not bias-corrected, in contrast to the procedure employed with the offline SimStrat-UoG model simulation. Although the spatial resolution of the SimStrat-UoG simulation is twice that of CLM5, the bias caused by its model deficiency is even higher. Clearly, when compared with SimStrat-UoG model (r value = 0.77; Fig. S9), CLM5 has better prediction skill (r value = 0.93; Fig. S1) in ice thickness variance, although the SimStrat-UoG model does not have the trend of underestimation towards thick ice cover.]. Therefore, we are confident in the robustness of our results.

Our lake model is fully coupled with a snowfall model which has implications for the heat transfer, radiation absorption, reflection etc., and it integrates with the lake ice growth. The cooling effect of cloud cover is also included in our simulation. In addition, our fully coupled model simulation allows us to resolve

diurnal cycle variations of radiation changes, thus simulating the cooling effect better than the current suite of offline lake simulations. Although snowfall and cloud cover certainly contribute to the changes in lake ice, surface warming and albedo feedback are the first order drivers of the ice changes. In our study, we consider changes in snowfall and cloud cover part of anthropogenic-induced climate change driving changes in ice phenology. A detailed assessment of the projected changes in snowfall and cloud cover is, however, beyond the scope of our manuscript.

In summary, to provide a comprehensive view of our simulations, we add a thorough discussion on the factors influencing lake ice [We believe this work is an important step forward in modeling lake ice and its response to greenhouse warming. Nevertheless, several caveats should be noted when interpreting our results. Our simulations use a gridded simulation method. Each 1° longitude-latitude grid cell has one representative lake whose mean depth is determined by area-weighted average of depths of all lakes in the grid cell. However, individual lakes within the grid cell could have lake ice dynamics different from the representative lake. This is due to, for example, elevation gradient and depth². Lake depth determines thermal inertia and mixing regimes of the water column, thereby impacting freezing and thawing processes of lake ice cover. Deeper lakes tend to take longer to freeze than shallower lakes under the same climate forcing. Moreover, the representative lakes were simulated with a one-dimensional lake model. Horizontal heterogeneity in water temperature and ice cover was not simulated. This is important for large lakes in which water circulation and ice cover mechanics in the horizontal direction impact lake ice phenology. Longer fetches in large lakes also lead to later freeze and earlier breakup by increasing wind speed, thus lacking horizontal features is likely to result in unrealistic ice phenology particularly in large lakes. However, the simulation of the representative lake does reflect changes in the most common lake type within the grid cell. Among all the factors like depth and lake size, air temperature is the dominant factor inducing lake ice changes according to a comprehensive analysis based on global observations², and these insufficiencies in one-dimensional models mainly impact the climatological mean more than variability of lake ice phenology. Long observation records from three lakes in one grid cell (1° x 1°) in Finland (Fig. S8) show large differences in their climatological means of ice duration, but their long-term changes are similar and captured well by the representative lake simulation. Although the simulation tends to have larger bias in climatological mean ice duration in large lakes, such as Lake Baikal (38.5 days) and Lake Superior (17.8 days) (Fig. S8), long-term changes, i.e., lake ice loss, over the last century are captured reasonably well by our simulation. It should also be noted that lake ice growth is sensitive to the timing and depth of snow on ice surface¹⁰, through the influence on ice thickness as well as altered albedo. While a 5-layer snow module is included in our model, uncertainty of snowfall derived from the atmospheric model could lead to inaccurate lake ice simulations. However, the full coupling between lakes and the atmosphere in our simulation does represent the most advanced modelling practice for lake ice dynamics to date by simulating the three-way interactions between water, ice and air and by resolving the diurnal cycle of lake thermodynamics.], and how our simulation advances current lake model simulations. We also describe the caveats of our simulation.

Grant, L. et al. Attribution of global lake systems change to anthropogenic forcing. *Nat Geosci*, doi:10.1038/s41561-021-00833-x (2021).

Key Results and Significance

There is some statistical analysis of trends in lake ice since the 1980s. Some more analysis and direct comparison to other periods of warming earlier in the 20th Century (i.e. how does recent lake ice loss compare to Early Twentieth Century Warming period?) would add substantial weight to the analysis of anthropogenic forcing and identification of emergent trends. Furthermore, direct comparison and discussion of model runs on an SSP1-1.9 with SSP3-7.0 would allow a clearer identification of anthropogenic forcing on future lake ice trends. Currently a more fitting title would be 'Analysis of Lake Ice Trends under High Emissions Climate Scenario (SSP3-7.0)' as analysis of past anthropogenic forcing on lake ice changes is limited (see comments) and comparison to model runs from a low emissions scenario in future would allow further clear identification of anthropogenic forcing.

In recognition of the reviewer's valid comments, we add related material [Our CESM2-LE simulations show that the first anthropogenic deviation from the natural ice phenology range occurred already between 1980-1990 (Fig. 5b), after which lake ice loss rates are the fastest in terms of ice duration over the last century in both simulations (Fig. S7b) and continuous observations from 30 lakes in the Northern Hemisphere¹⁶. In comparison, ice loss rates were insignificant in the first half of 20th century when the impact of anthropogenic forcings was minimal (Fig. S7a).] and Fig. S7 to illustrate trends of ice duration in 1900-1930 and 1980-2010, respectively. While lake ice loss is insignificant in 1900-1930 in both, simulations (Fig. S7) and the continuous observations (Fig. 3C in Sharma et al., 2021), it is very pronounced from 1980-2010. This underlines the fact that the anthropogenic forcing certainly has already generated lake ice loss on a global scale over recent decades. This is consistent with our Time of emergence (TOE) analysis, i.e., the anthropogenic induced ice loss emerged since 1970s. Our TOE calculation using the trend versus variability ratio includes this information. In 1850-1950, the forced perturbation is minimum, with the variance in individual members primarily induced by the natural variability. As a result, the ratio of the forced trend to natural variability is small, and the anthropogenic signal did not emerge. After 1970, the lake ice loss trend grew in amplitude and emerged from the natural background variability, implicating greenhouse warming as the driver of ice loss.

It would be ideal to compare the projections under different scenarios. However, to our knowledge, there is no large ensemble simulation available for lakes except our work. To put the computational demand of our simulations into perspective, the 100 SSP370 ensemble members consumed nearly a year of South Korea's 2nd fastest academic supercomputer. Given the enormous computing costs, it will be prohibitive in the immediate future to rerun this fully coupled model simulation for lower-emission SSP scenario. The ISIMIP-2b project is the only publicly available suite of simulations for lakes which was conducted using offline multi-model ensembles and different scenarios. However, these offline simulations cannot capture natural variability which makes it difficult to determine TOE for anthropogenic signals and no-analogue conditions in lake ice for individual lake grid cells.

In order to provide a more comprehensive view of lake ice loss under different warming levels, we have now compared the TOE of the anthropogenic signal and no-analogue conditions to global mean surface air temperature anomalies (modifying Fig. 5, adding Fig. S6 and Lines 286-293) by following the recommendations from the Intergovernmental Panel on Climate Change (IPCC)'s sixth Assessment report. Since lake ice duration responds to surface warming linearly in different scenarios (Fig. 4f in Grant et al.,

2021), our results can provide an important reference for TOE analysis under different forcing scenarios. For example, no-analogue conditions start to emerge under 1.9 °C warming based on our simulations. Under SSP370, no-analogue conditions start to emerge in 2040. Under a low emissions scenario, we can estimate that no-analogue conditions may start to emerge in 2060 when global warming reaches a 1.9 °C warming level. Of course, for more robust quantitative estimations it would be necessary to compare lake ice loss under different forcing scenarios in the future when more large ensemble simulations become available.

Sharma, S., Richardson, D. C., Woolway, R. I., Imrit, M. A., Bouffard, D., Blagrove, K., et al. (2021). Loss of ice cover, shifting phenology, and more extreme events in Northern Hemisphere lakes. *Journal of Geophysical Research: Biogeosciences*, 126, e2021JG006348. <https://doi.org/10.1029/2021JG006348>
Grant, L. et al. Attribution of global lake systems change to anthropogenic forcing. *Nat Geosci*, doi:10.1038/s41561-021-00833-x (2021).

Beyond this there are some substantial findings, particularly regarding predictions of lake ice loss in the Canadian Arctic and Tibetan plateau. The former is logically argued to be associated with loss of sea ice in the Arctic and consequent Arctic amplification of climate change during the 'cold season', which in reality will be particularly important during the Autumn (please be more specific than 'the Fall'). Given this important result and identification of feedback mechanisms, it is also evident from the figures that areas in the Russian Arctic surrounding the Barents sea sector have experienced lake ice loss too, which should be noted in the text. The identification of areas such as the Tibetan plateau that are predicted to have substantial further lake ice loss, and experience non-analogous ice coverage within the next 4-5 decades are important results, for ecology and also indigenous populations that may travel over them during winter.

We corrected the statement about the Russian Arctic surrounding the Barents accordingly (Lines 144, 177, and 186).

Validity, Data and Methodology

The data and methodology are of a fairly high standard, particularly with the validation data. However, just using one model could be questioned. Although there is some comparison to other models. I am not a modeller so other reviewers comments on this should be favoured.

We agree that a comparison with different lake models would be informative.

We compare our simulation with the SimStrat-UoG model simulation (Lines 442-462) and demonstrate that CESM2 has a comparable and in some areas even better performance than SimStrat-UoG simulation for lake ice. We therefore consider our fully coupled simulation robust relative to other available lake model simulations. Our study is the first of its kind to explicitly account for natural climate variability (using 90 ensemble members) and full coupling. In future, we hope that in the future there will be more inter-model comparison with similar types of models.

Analytical Approach

Analytical approach seems to be fairly sound. Although again I am not a modeller so cannot comment fully on this.

Thank you.

Suggested Improvements

There should be more of an introduction to the main factors affecting lake ice loss and an overview of how/which of these are parameterised in the model, which should be accessible to non-specialists. Several sections of the text should be reviewed so that the meaning is clearer for non-modeller scientists. Some restructuring to help build the argument in stages for the reader and not get lost in the modelling detail would substantially improve the accessibility of the article.

We thank the reviewer for the suggestions.

We have added a thorough discussion [We believe this work is an important step forward in modeling lake ice and its response to greenhouse warming. Nevertheless, several caveats should be noted when interpreting our results. Our simulations use a gridded simulation method. Each 1° longitude-latitude grid cell has one representative lake whose mean depth is determined by area-weighted average of depths of all lakes in the grid cell. However, individual lakes within the grid cell could have lake ice dynamics different from the representative lake. This is due to, for example, elevation gradient and depth². Lake depth determines thermal inertia and mixing regimes of the water column, thereby impacting freezing and thawing processes of lake ice cover. Deeper lakes tend to take longer to freeze than shallower lakes under the same climate forcing. Moreover, the representative lakes were simulated with a one-dimensional lake model. Horizontal heterogeneity in water temperature and ice cover was not simulated. This is important for large lakes in which water circulation and ice cover mechanics in the horizontal direction impact lake ice phenology. Longer fetches in large lakes also lead to later freeze and earlier breakup by increasing wind speed, thus lacking horizontal features is likely to result in unrealistic ice phenology particularly in large lakes. However, the simulation of the representative lake does reflect changes in the most common lake type within the grid cell. Among all the factors like depth and lake size, air temperature is the dominant factor inducing lake ice changes according to a comprehensive analysis based on global observations², and these insufficiencies in one-dimensional models mainly impact the climatological mean more than variability of lake ice phenology. Long observation records from three lakes in one grid cell (1° x 1°) in Finland (Fig. S8) show large differences in their climatological means of ice duration, but their long-term changes are similar and captured well by the representative lake simulation. Although the simulation tends to have larger bias in climatological mean ice duration in large lakes, such as Lake Baikal (38.5 days) and Lake Superior (17.8 days) (Fig. S8), long-term changes, i.e., lake ice loss, over the last century are captured reasonably well by our simulation. It should also be noted that lake ice growth is sensitive to the timing and depth of snow on ice surface¹⁰, through the influence on ice thickness as well as altered albedo. While a 5-layer snow module is included in our model, uncertainty of snowfall derived from the atmospheric model could lead to inaccurate lake ice simulations.] in the manuscript about factors influencing lake ice, as well

as advantages and disadvantages of the parameterization in the model, to provide an objective interpretation for readers. We have also revised the manuscript according to the suggestions.

Clarity and Context

Generally reasonably good. The language needs to be changed in some sections, for results regarding predictions of future outcomes there is uncertainty, so 'will' should be changed to 'may'. Please avoid referring to 'the fall' and refer to seasons (autumn) and ideally specific months.

Thank you very much.

We revised the manuscript accordingly.

REVIEWER COMMENTS

Reviewer #1 (Remarks to the Author):

I thank the authors for their response to my comments and questions. I think they've appropriately placed their study in context with other analyses of lake ice projections, and certainly this work provides an improved approach through the use of a coupled system (e.g., CESM). I do think at hand this manuscript represents a well crafted approach, although limited as a case study of one climate scenario. This certainly leaves room for future research to build upon, as well as investigation of how 1D lake models perform along coastal regions where ice forms and where three-dimensional processes are critical, but I think the manuscript is suitable as it stands. I have no further comments.

Reviewer #2 (Remarks to the Author):

Overview

The authors have produced a substantially improved manuscript, which provides an important advance on model predictions of future changes in lake ice conditions in light of Anthropogenic forcing through the greenhouse effect. The article gives detailed insight into the main predictions for general trends in lake ice cover and thickness, with some detailed analysis of regional patterns around the globe and the forcings behind them, nicely illustrated by figure 5.

Key Results and Significance

A series (~90) of model runs have been conducted with the same model, which crucially incorporates lake-ice-air interactions, unlike previous statistically based models. This, combined with resolving diurnal cycle thermodynamics (30 minute resolution) represent a substantial advancement in the study of lake ice phenology, as well as important predictions of associated ecosystems shifts. The authors argue that the spread of these runs captures natural variability and that the mean represents anthropogenic forcing. Whilst this keeps consistency in the physical parameters within the model it obviously relies on just one model capturing physical processes, which the authors acknowledge is a limitation but highlight it performs as well if not better than other competing models. Furthermore, there is good validation of the model from Canadian lake data. Although it should also be noted that lake temperatures and ice phenology response to climate can be heterogenous, which they discuss and do a good job of incorporating into a global dataset (which will always involve compromises). They now use the opportunity to call for other integrated lake-climate models to be developed in order to push the science forward. This is a particularly important point, as the authors highlight warming needs to be kept below 1.9oC to avoid widespread non-analogous changes in lake ice phenology. This could also be added into the title.

There is now very good discussion of parameterisation of key factors that may affect lake ice phenology; such as changes in snowfall (potentially increasing ice thickness etc) and wind patterns/speeds. This is of high importance for driving the science forward and highlighting important areas for future research, such as improving predictions of lake ice freeze up. They acknowledge that in reality the conditions controlling lake ice freeze up are highly variable. Greater agreement between researchers on a critical lake ice threshold (0cm seems too thin) may provide an important step towards further advancing the research field in selecting a suitable ice thickness threshold (5 to 10cm seems more realistic) beyond which lake ice is highly resistant to meteorological conditions and therefore more likely to persist.

There is now detailed statistical analysis of trends in lake ice since the 1980s and direct comparison to other periods of warming earlier in the 20th Century. This adds substantial weight to the analysis of Anthropogenic forcing and identification of emergent trends. The authors acknowledge that comparison to model runs from a low emissions scenario in future would allow further clear identification of Anthropogenic forcing, but is computationally prohibitive and another important area of further research.

There are some substantial findings from regional, particularly regarding predictions of lake ice loss in the Canadian Arctic, Barents sea area and Tibetan plateau. The former is logically argued to be associated with loss of sea ice in the Arctic and consequent Arctic amplification of climate change

during the 'cold season', which in reality will be particularly important during the Autumn. The identification of areas such as the Tibetan plateau that are predicted to have substantial further lake ice loss, and experience non-analogous ice coverage within the next 4-5 decades are important results, for ecology and also indigenous populations that may travel over them during winter.

Validity, Data and Methodology

The data and methodology are of a high standard, particularly with the validation data and comparison to other models. The model runs for such an advanced model are computationally demanding and a substantial undertaking. A comment in the methodology on how long these took to run would not be a miss! I am not a modeller so other reviewers comments on this should be favoured.

Analytical Approach

Analytical approach seems to be fairly sound. Although again I am not a modeller so cannot comment fully on this.

Suggested Improvements

The manuscript has now been substantially improved so that the article is easily accessible to non-modellers and the novelty of the research is fairly well communicated. To further do justice to the novelty of the research I would just suggest some very minor amendments;

51 'We address these limitations here by...'

77 'Thereby addressing the limitations of...'

The title should be changed to 'Emerging Patterns of Anthropogenic Lake Ice Loss' as 'Emergence' has ecological connotations of systematic interactions. The argument of the article is that there are large perturbations in systems, which have exhibited some heterogeneity to human forcing of the climate system through greenhouse effect. They could also add something like 'and Predicted Ice Phenology Thresholds for Ecosystem Shifts' to do full justice to the novelty of the article (but not essential).

Clarity and Context

Generally very good and now accessible to non-modellers. It is now an intriguing read.

n) and ideally specific months.

REVIEWERS' COMMENTS

Reviewer #1 (Remarks to the Author):

I thank the authors for their response to my comments and questions. I think they've appropriately placed their study in context with other analyses of lake ice projections, and certainly this work provides an improved approach through the use of a coupled system (e.g., CESM). I do think at hand this manuscript represents a well crafted approach, although limited as a case study of one climate scenario. This certainly leaves room for future research to build upon, as well as investigation of how 1D lake models perform along coastal regions where ice forms and where three-dimensional processes are critical, but I think the manuscript is suitable as it stands. I have no further comments.

Thank you very much!

Reviewer #2 (Remarks to the Author):

Overview

The authors have produced a substantially improved manuscript, which provides an important advance on model predictions of future changes in lake ice conditions in light of Anthropogenic forcing through the greenhouse effect. The article gives detailed insight into the main predictions for general trends in lake ice cover and thickness, with some detailed analysis of regional patterns around the globe and the forcings behind them, nicely illustrated by figure 5.

Key Results and Significance

A series (~90) of model runs have been conducted with the same model, which crucially incorporates lake-ice-air interactions, unlike previous statistically based models. This, combined with resolving diurnal cycle thermodynamics (30 minute resolution) represent a substantial advancement in the study of lake ice phenology, as well as important predictions of associated ecosystems shifts. The authors argue that the spread of these runs captures natural variability and that the mean represents anthropogenic forcing. Whilst this keeps consistency in the physical parameters within the model it obviously relies on just one model capturing physical processes, which the authors acknowledge is a limitation but highlight it performs as well if not better than other competing models. Furthermore, there is good validation of the model from Canadian lake data. Although it should also be noted that lake temperatures and ice phenology response to climate can be heterogenous, which they discuss

and do a good job of incorporating into a global dataset (which will always involve compromises). They now use the opportunity to call for other integrated lake-climate models to be developed in order to push the science forward. This is a particularly important point, as the authors highlight warming needs to be kept below 1.9oC to avoid widespread non-analogous changes in lake ice phenology. This could also be added into the title. There is now very good discussion of parameterisation of key factors that may affect lake ice phenology; such as changes in snowfall (potentially increasing ice thickness etc) and wind patterns/speeds. This is of high importance for driving the science forward and highlighting important areas for future research, such as improving predictions of lake ice freeze up. They acknowledge that in reality the conditions controlling lake ice freeze up are highly variable. Greater agreement between researchers on a critical lake ice threshold (0cm

seems too thin) may provide an important step towards further advancing the research field in selecting a suitable ice thickness threshold (5 to 10cm seems more realistic) beyond which lake ice is highly resistant to meteorological conditions and therefore more likely to persist.

There is now detailed statistical analysis of trends in lake ice since the 1980s and direct comparison to other periods of warming earlier in the 20th Century. This adds substantial weight to the analysis of Anthropogenic forcing and identification of emergent trends. The authors acknowledge that comparison to model runs from a low emissions scenario in future would allow further clear identification of Anthropogenic forcing, but is computationally prohibitive and another important area of further research.

There are some substantial findings from regional, particularly regarding predictions of lake ice loss in the Canadian Arctic, Barents sea area and Tibetan plateau. The former is logically argued to be associated with loss of sea ice in the Arctic and consequent Arctic amplification of climate change during the 'cold season', which in reality will be particularly important during the Autumn. The identification of areas such as the Tibetan plateau that are predicted to have substantial further lake ice loss, and experience non-analogous ice coverage within the next 4-5 decades are important results, for ecology and also indigenous populations that may travel over them during winter.

Thank you very much for the constructive comments!

Validity, Data and Methodology

The data and methodology are of a high standard, particularly with the validation data and comparison to other models. The model runs for such an advanced model are computationally demanding and a substantial undertaking. A comment in the methodology on how long these took to run would not be a miss! I am not a modeller so other reviewers comments on this should be favoured.

Yes, it takes a whole year to run this project on a supercomputer on the IBS/ICCP supercomputer "Aleph" 1.43-petaflop high-performance Cray XC50-LC Skylake computing system with 18,720 processor cores, 9.59-petabyte storage. And we include this in the acknowledgement.

Analytical Approach

Analytical approach seems to be fairly sound. Although again I am not a modeller so cannot comment fully on this.

Thank you very much!

Suggested Improvements

The manuscript has now been substantially improved so that the article is easily accessible to non-modellers and the novelty of the research is fairly well communicated. To further do justice to the novelty of the research I would just suggest some very minor amendments;

51 'We address these limitations here by...'

77 'Thereby addressing the limitations of...'

We edit it accordingly in the text (Line 47-48: We address these limitations here by including lake simulators into Earth system models as an interactive component).

The title should be changed to 'Emerging Patterns of Anthropogenic Lake Ice Loss' as 'Emergence' has ecological connotations of systematic interactions. The argument of the article is that there are large perturbations in systems, which have exhibited some heterogeneity to human forcing of the climate system through greenhouse effect. They could also add something like 'and Predicted Ice Phenology Thresholds for Ecosystem Shifts' to do full justice to the novelty of the article (but not essential).

By following this comment and the comment from editor, we now change the title to "Emerging unprecedented lake ice loss in climate change projections", which is clear and fit the topic of this study more closely.

Clarity and Context

Generally very good and now accessible to non-modellers. It is now an intriguing read.

Thanks again for your constructive comments.